# Effectiveness of Low-Level Laser Therapy with a 915 Nm Wavelength Diode Laser on the Healing of Intraoral Mucosal Wound: An Animal Study and a Double-Blind Randomized Clinical Trial

**DOI:** 10.3390/medicina55080405

**Published:** 2019-07-24

**Authors:** Han-Wool Choung, Sung-Ho Lee, Ahe Reum Ham, Na Ra Lee, Bongju Kim, Kang-Mi Pang, Jeong Won Jahng, Jong-Ho Lee

**Affiliations:** 1Department of Oral and Maxillofacial Surgery, Seoul National University Dental Hospital, Seoul 03080, Korea; 2Dental Life Science Research Institute & Clinical Translational Research Center for Dental Science, Seoul National University Dental Hospital, Seoul 03080, Korea; 3School of Dentistry and Dental Research Institute, Seoul National University, Seoul 03080, Korea; 4Clinical Research Team, Dentium, 21, Seoul 06169, Korea; 5Department of Oral and Maxillofacial Surgery, Seoul National University Kwanak Dental Hospital, Seoul 08826, Korea

**Keywords:** laser dentistry, low-level laser therapy, 915 mm diode laser, mucosal wound healing

## Abstract

*Background and objectives:* Diode laser has been the most popular low-level laser therapy (LLLT) technique in dentistry due to its good tissue penetration, lower financial costs, small size for portable application, and convenience to use. A series of recent studies with 940 nm or 980 nm lasers demonstrated that LLLT showed positive effects after third molar extraction or periodontal flap surgery. However, the effects of LLLT on intraoral mucosal wound healing after surgical incision have not yet been determined in human clinical study. *Materials and Methods:* The present study was performed to determine the efficacy and safety of 915 nm wavelength low-level laser therapy (LLLT) in mucosal wound healing. A total of 108 Sprague–Dawley rats were used. They were divided into three groups: Abrasive wound group, immediate LLLT once group, and daily LLLT group. As a clinical study, a total of 16 patients with split-mouth design subjected to bilateral mandibular third molar extraction were allocated into the LLLT group and placebo group. The process of LLLT was performed on postoperative days 0, 1, and 7, and parameters related to wound healing were analyzed on days 1, 7, and 14. *Results:* Repeated laser irradiation promoted mucosal wound healing of the rats. In the clinical study, although there were no significant statistical differences between the LLLT and placebo groups in all inflammatory parameters, the early stage mucosal healing tendency of wound dehiscence was higher in the LLLT group than in the placebo group clinically on postoperative day 1. *Conclusions:* The present results showed that 915 nm LLLT could be applied safely as an auxiliary therapy for mucosal wound healing.

## 1. Introduction

Surgical extraction of third molar is one of the most common oral surgeries in dentistry. After extraction, a traumatic inflammatory response occurs along the mucosal wound with pain and facial swelling [1]. The mucosal wound remains under high risk of infection in the oral environment [2]. To reduce the postoperative complications and promote healing of the mucosal wounds, application of additional methods could be considered [3]. Low-level laser therapy (LLLT) is increasingly being used for reducing postoperative complications after various surgical procedures in intraoral regions because it enhances tissue regeneration and wound healing, as well as decreasing pain and edema through anti-inflammatory procedures [4]. This anti-inflammatory procedure is achieved by a mechanism of increased adenosine triphosphate (ATP) synthesis and decreased oxidative stress [3,5,6]. The accelerated healing effect of laser is caused by stimulation of natural biological processes [7]. Furthermore, the advantageous effects of LLLT were also reported by several in vitro experiments; LLLT induces proliferation of human gingival fibroblast [8] and reduces inflammatory conditions [9].

Previous studies, however, showed debatable results regarding the efficacy of LLLT [10]. Batinjan et al. [11] have proven that LLLT significantly reduced postoperative complications after surgical removal of third molars. On the other hand, it has been documented that no statistically significant differences were observed in the levels of pain, postoperative swelling, or trismus between the control and LLLT groups after a third molar extraction [8]. One of the reasons for this controversy is that the conditions of optimal laser irradiation for anti-inflammatory and accelerated healing effects after oral surgery have not been established due to its diverse forms such as output power, energy, duration, pulse rate, and wavelength of laser.

Recently, diode laser has been the most popular LLLT technique in dentistry for its good tissue penetration, lower financial costs, small size for portable application, and convenience to use [12]. In LLLT attempts in the intraoral area, diode laser with a wavelength between 655 and 980 nm have been used, which is able to promote healing process on the extraction site and to induce angiogenesis [13,14]. A series of recent studies with 940 nm or 980 nm lasers demonstrated that LLLT showed positive effects after third molar extraction or periodontal flap surgery [15,16,17,18]. However, the effects of LLLT on intraoral mucosal wound healing after surgical incision have not yet been determined in human clinical study.

Given these findings, the present study aimed to examine the efficacy and safety of 915 nm wavelength laser irradiation on mucosal healing after extraction of mandibular third molars with surgical incisions, and to serve as a pilot study offering basic data for a future confirmatory clinical trial of intraoral wound healing. To attain more convincing scientific evidence of the effects of 915 nm LLLT on mucosal wound healing, a rat animal model study was preceded. After obtaining promising results from the pre-clinical animal study, the effects of LLLT on humans were evaluated as a clinical study.

## 2. Materials and Methods

### 2.1. Animal Study 

To evaluate the effect of 915 nm LLLT on mucosal wound healing in vivo, a total of 108 rats (Sprague–Dawley rats, 7-weeks-old, male, weight 250–300 g, Orient Bio, Gapyeong, Republic of Korea) were used for this experiment, and they were randomly divided into three groups: an abrasive wound-induced group as a negative control (Control group), an immediate 915 nm laser irradiation subjective to abrasive wound group (LLLT once group), and a daily 915 nm laser irradiation subjective to abrasive wound group (LLLT daily group). Each group consisted of 6 animals per day for 6 days, totaling 36 rats per group (Table 1).

### 2.2. Surgical Procedures

All experiments using animals followed protocols approved by the Institutional Animal Care and Use Committee of Seoul National University (SNU-151116-5-2) and the principles of 3Rs (replacement, refinement, or reduction) to prevent injuries and to promote efficiency and management. The animals were bred in a specific-pathogen-free laboratory at a constant temperature (21 ± 1 °C) and humidity (50%) with 12 h light–dark cycle (light; 07:30–20:00, dark; 20:00–07:30). Regular food (Purina Rodent Chow, Purina Co., Seoul, Republic of Korea) and purified water were provided ad-libitum, with a quarantine period of one week. After the adjustment period, a 3 mL mixed solution (100 mg/kg) of pentobarbital (Hanlim Pharm, Co., LTD, Gyeong-gi, Republic of Korea) and chloral hydrate (Sigma-Aldrich, Co., On, Canada) was abdominally injected into the rats for anesthesia. To produce an intraoral mucosal wound, a round diamond bur with 15,000 round per minute (rpm) speed was used to induce an abrasion wound on the palatal mucosa creating a 1 × 1 cm sized wound. Subsequently, LLLT (Dental 5, Bison Medical Co., Seoul, Korea) once and LLLT daily groups were progressed in the following settings: 915 nm in wavelength, 0.2 W in output power, 150 J/cm^2^ in fluency, and 0.25 W/cm^2^ in power density, in the form of single point Gaussian beams and at continuous mode with 1 cm distance for 10 min. All animals were subjected to euthanasia using carbon dioxide. The wounds were closely observed and evaluated for the healing progress, and the wound areas were measured using ImageJ software version 1.8.0 (National Institutes of Health, Bethesda, MD, USA) for 1 to 6 days. Finally, 7 days after abrasion on the mucosa, the specimens were harvested, and serial 5 μm thick sections were cut and stained with hematoxylin/eosin (H/E) for histological analysis.

### 2.3. Clinical Study Participants

As an exploratory clinical trial (interim analysis), all data were collected from 16 patients of Seoul National University Dental Hospital (Seoul, Korea) who underwent surgical extraction of impacted third molar on each side of the mandible. Clinical Information Research Service (CRIS: PRE20180814-005, Registered 14 August 2018), The Ministry of Food and Drug Safety (MFDS: A37010.10, Registered 8 September 2016), and Institutional Review Board (IRB: CDE 16009, approved 20 September 2016) approved the study protocol. All of the data were collected in the Clinical Trial Center of Seoul National University Dental Hospital.

### 2.4. Inclusion/Exclusion Criteria

The inclusion criteria were as follows: Indications for surgical removal of both mandibular third molars with partial impaction which required surgical incision, similar vertical and horizontal positions of both sides of the third molars, and aged above 19 years. This study excluded patients with systemic conditions that contraindicated the surgical procedure, fully erupted mandibular third molars which do not need surgical incision, acute pericoronitis of third molars, generally poor oral hygiene and conditions, pregnant or nursing women, smokers, those under medications that would interfere with wound healing, or patients who refused to give the signed informed consent. None of the subjects had undergone a similar treatment previously to that performed in this study.

### 2.5. Clinical Study Procedures

All surgical extractions were performed by two experienced oral and maxillofacial surgeons (OMFS) following a standardized protocol so as not to influence the surgical trauma level by surgical experience of the operator. Before surgery, 2 ampoules of 2% lidocaine with 1:100,000 epinephrine (2% Lidocaine HCL Injection, Huons Co., Ltd., Seongnam, Korea) were injected for local anesthesia. Similar degree and size of mucoperiosteal incisions (about 1 cm) were made on solid bony surfaces on both sides of the mandible. The third molars were removed after elevation of mucoperiosteal flap and odontomy. After extraction, the mucosal incision sites were sutured using 4-0 Dafilon (B. Braun, Melsungen AG, Melsungen, Germany). For medication, the patients were administered 250 mg of clarithromycin 30 min before surgery, and 250 mg clarithromycin, 100 mg aceclofenac, and 1 tab of phazyme twice after the extraction on the day of surgery. During the study period, the patients were not allowed to take any non-prescribed medication.

All participants visited the clinic 5 times in total: On the day of screening (Visit 1), on the day of extraction (Visit 2), and on postoperative days 1 (Visit 3), 7 (Visit 4), and 14 (Visit 5). On the screening day, the patients were examined by an interview and panoramic X-rays after completing the written consent form. Based on this examination, the subjects eligible for this clinical trial were selected. All 16 patients fulfilled the participants criteria. On the day of extraction, a total of 32 cases (split-mouth design: Right and left sides of the mandibles of 16 participants) were randomly assigned to be in the experimental group (n = 16) or the placebo control group (n = 16) by parallel design using Excel software.

On the day of extraction (after extraction) and on postoperative days 1 and 7, the experimental group received 6 sets of 50 s long LLLT (Dental 5, Bison Medical Co., Seoul, Korea) with settings of 915 nm wavelength, 0.5 W power, 187.5 J/cm^2^ in fluency, and 0.625 W/cm^2^ in power density, in the form of single point Gaussian beams, continuous mode, 1 cm distance from the laser tip to the mucosal wound, and 0.8 cm^2^ of beam area on the mucosal incision lines by a ‘clinical trials coordinator’. LLLT does not produce any heat nor damage any tissue. The placebo group underwent the same treatment with a laser probe that was not turned on. Therefore, the participants were blinded and were not able to recognize whether they received treatment of the placebo or experimental group. After the process, an ‘experienced evaluator’ assessed the degree of wound healing of mucosal incisions in the experimental and control groups and identified any concomitant medications or adverse cases. The subjects were blinded during the study, and separate evaluators, excluding the operators, assessed the degree of wound healing of the extraction sites. After the sutures were removed on the postsurgical day 14, an evaluator assessed the degree of wound healing and identified any concomitant medications or adverse cases.

In analysis of the degree of wound healing, this study referred to the method of Simunovic et al. [19]. In total, 4 parameters were scored: redness, swelling, pain, and incision wound closure. Each parameter was divided into 4 ratings (asymptomatic, mild, moderate, and severe) and scored from 0 to 15 (Table 2, score 0 indicating ‘no symptom’ to score 15 indicating ‘severe’ and maximum severity). Expertly skilled evaluators analyzed the summed values of all parameters and separately examined the value for each parameter. Adverse cases were recorded each visit according to the following criteria: Occurrence date, disappearance date, response ratings (mild, moderate, and severe), and abnormal case results.

### 2.6. Statistical Analysis

Statistical analysis of the results was based on the ‘Guidelines for Clinical Trial Statistics’ (KFDA: Korea Food and Drug Administration, 2000), and all data were analyzed using SAS software (Version 9.4, SAS Institute, Cary, NC, USA). All values were presented as mean ± standard deviation (SD), and the applied statistical significance level was 5%. Repeated measure ANOVA analysis was used to compare the degree of wound healing of the LLLT and placebo groups on each visit. The categorical data were analyzed by Fisher’s exact test, and adverse cases reported during the study were analyzed through the Chi-square test and Fisher’s exact test.

## 3. Results

### 3.1. Animal Study

To examine the effect of 915 nm LLLT on mucosal wound healing in vivo, a preclinical animal study was performed under three different conditions: Mucosal wound without LLLT (Control), mucosal wound with immediate LLLT once, and mucosal wound with daily LLLT. All of the surgical wounds recovered within seven days, clinically, without any adverse event (Figure 1A). ImageJ software was used to analyze the wound healing by measuring the size of wound (Figure 1B). As a result, the size of wound was significantly decreased by daily LLLT on the third and fourth days. These findings suggest that daily LLLT might induce early stage wound healing of intraoral mucosa.

Based on the results of clinical examination of the wound size, the histology of the mucosal wounds was evaluated under H/E staining. Seven days after producing the wound, the changes in the epithelial layer among three groups were analyzed. The control group showed that the epithelial layers of the wound site were thicker than those of normal tissue after seven days of healing period (Figure 2A). In the LLLT once group, the thickness of the epithelial layers of the wound site were still irregular, similar to the control group although they were less thick than those of the control group (Figure 2B). On the other hand, the daily LLLT group was found to have thin uniform epithelial layers of the wound site similar to that of normal tissue (Figure 2C).

To sum up, these in vivo findings indicate that irradiation of 915 nm wavelength diode laser promotes intraoral mucosal wound healing clinically and histologically, and repeated irradiation of 915 nm laser shows better results than a single irradiation.

### 3.2. Clinical Study

All 16 patients who underwent surgical extraction of impacted third molar on each side of the mandible between September 2016 and March 2017 agreed to and signed the informed consent to participate in the study. The age of the 16 participants ranged from 18 to 42 years old (average 25.06 ± 7.12 years). Among them, eight were females and eight were males. There were no specific premedical histories except; one subject had an allergic specific constitution within one year, 5 patients had a history of health problems, including temporomandibular joint syndromes, osteodystrophy, fibrous dysplasia of the jaw, neurofibromatosis, and malocclusion. All 32 cases were successfully analyzed without excluding any participant. Detailed efficacy evaluation was derived from scores for each parameter including redness, swelling, pain, and dehiscence of the incisional wound. Overall efficacy evaluation was based on the sum of the four clinical parameters.

#### 3.2.1. Redness and Swelling Intensities

In redness intensity, the LLLT group scored 6.44 ± 2.63 on day 1, 2.13 ± 2.42 on day 7, and 0.19 ± 0.75 on day 14; the control group scored 7.06 ± 2.52 on day 1, 2.50 ± 2.19 on day 7, and 0.00 ± 0.00 on day 14 (Figure 3A). In swelling intensity, the LLLT group scored 6.63 ± 2.19 on day 1, 1.88 ± 1.71 on day 7, and 0.13 ± 0.50 on day 14; the control group scored 7.19 ± 2.66 on day 1, 3.00 ± 2.48 on day 7, and 0.00 ± 0.00 on day 14 (Figure 3B). There were significant differences in redness and swelling scores according to time of visit (*p* < 0.05), but the degrees of the difference between the two groups were not significant (*p* > 0.05).

#### 3.2.2. Pain Intensity and Dehiscence of the Wound

In pain intensity, the LLLT group scored 4.69 ± 3.46 on day 1, 2.56 ± 2.99 on day 7, and 0.63 ± 1.71 on day 14; the control group scored 5.75 ± 4.19 on day 1, 3.00 ± 3.10 on day 7, and 0.63 ± 1.71 on day 14 (Figure 4A). Regarding dehiscence of the wound, the LLLT group scored 5.25 ± 1.88 on day 1, 2.25 ± 2.65 on day 7, and 0.94 ± 1.81 on day 14; the control group scored 5.69 ± 2.70 on day 1, 2.31 ± 2.09 on day 7, and 0.69 ± 1.45 on day 14 (Figure 4B). There were significant differences in both scores according to time of visit (*p* < 0.05), but the degrees of the difference between the two groups were not significant (*p* > 0.05).

#### 3.2.3. Overall Efficacy Evaluation

The LLLT group scored 23.00 ± 8.07 on day 1, 18.81 ± 7.21 on day 7, and 1.88 ± 2.16 on day 14; the control group scored 25.69 ± 8.79 on day 1, 10.81 ± 6.68 on day 7, and 1.31 ± 2.18 on day 14 (Figure 5). Despite the numerical differences between the degrees of wound healing in the experimental and control groups, the distinction was not significant (*p* > 0.05).

#### 3.2.4. Safety Evaluation

There were no unpredictable or severe postoperative events in all groups. One patient (6.25%) in the LLLT group and three patients (18.75%) in the control group had adverse events. Abnormalities included one case of mild postoperative pain and swelling in the LLLT group, while two cases of mild postoperative pain and swelling, and one case of postoperative infection were observed in the control group (Table 3 and Table 4). One patient showed postoperative pain and swelling on both control and LLLT sides. All adverse event cases were related with alveolar osteitis, so they were predictable, milder than severe, and had recovered during the follow-up visits.

## 4. Discussion

LLLT is based on its photo-biostimulation effects including promoting wound healing, anti-inflammatory effects, and reduction of pain by stimulating tissues without producing any irreversible changes [20]. These mechanisms are achieved by stimulation of natural biological processes, dose-dependent reduction of tumor necrosis factor-alpha concentration in acute phase of inflammation, change in size and permeability of vessel lumen, and alteration of neurotransmitter activity [7,21]. These efficacies of LLLT have been proven to accelerate dermal wound healing in the medical fields by numerous studies [22,23,24]. In contrast to the dermal tissue, the oral cavity is a special wet environment in which wound healing occurs in warm saliva [25]. The saliva contains numerous microorganisms and food debris, which might possibly act as a medium for infectious bacteria [26]. After surgical extraction of third molars, the incision wound in the oral mucosa remain under high risk of postoperative infection and inflammatory reactions [27]. Therefore, to prevent infection and promote healing of the mucosal wounds, we investigated the effect of 915 nm laser irradiation on mucosal wound healing in animal and human clinical studies.

Recently, lasers have been successfully used for clinical treatment of various intraoral diseases. It has been demonstrated that LLLT reduces size and related pain of oral lichen planus (OLP) and recurring aphthous stomatitis (RAS) [28,29]. Furthermore, lasers are being increasingly incorporated into conventional treatment for oral diseases as a part of surgical and postoperative therapy. Additional LLLT following root planing reduces gingival inflammation in chronic periodontitis [4]. Superpulsed LLLT reduces postoperative pain after extraction of a tooth [30]. However, the effect of LLLT with 915 nm wavelength diode laser on intraoral mucosal wound healing after surgical incision has not yet been reported.

In the present study, the in vivo preclinical findings indicate that 915 nm LLLT accelerated intraoral mucosal wound healing, and repeated irradiation of laser shows better results than a single irradiation. These results are consistent with those of the study which reported that repeated laser therapy after scaling showed better results compared to a single laser therapy after scaling group and scaling only group [31]. Multiple sessions of diode laser therapy showed a faster and greater tendency to reduce inflammatory responses. Such results might be explained by the cumulative effect of laser irradiation, by which every newly irradiated dosage postoperatively stays in the target tissue [11]. These findings from the animal study were reflected in the clinical study with a protocol of multiple irradiation of a 915 nm laser—0.5 W power, 187.5 J/cm^2^ in fluency, and 6 sets of 50 s long LLLT on the day of extraction and on postoperative days 1 and 7. To prevent excess heat production on the mucosa, a 1 min discontinuation was considered on the mucosal incision lines.

The limitation of the animal study is that the wounds were not identical between animal and clinical studies. The intraoral wound of animal was an abrasion wound, whereas the human wound was a mucoperiosteal incisional wound. Furthermore, intraoral mucosal thickness and healing potential are different between those two species. Therefore, we used some different parameters of 915 nm laser irradiation in animal and clinical studies because the wounds were not identical.

According to the overall analysis of efficacy, the LLLT and control groups showed significant changes with time, meaning that both groups healed over time as expected. However, the statistical difference between the two groups was not significant. Clinically, faster immediate wound healing was observed in the LLLT group than in the control group on postoperative day 1, although the degrees of the difference were not that significant in terms of *p*-value (*p* > 0.05). The analysis of each parameter displayed similar results: Severity figures for swelling and dehiscence of incision wound were all clinically better immediately in the LLLT group than in the control group. These results correspond well with those of the previous study with 940 nm LLLT on days 2 and 7 after extraction, which reported that there was less swelling and trismus in the LLLT group than in the control group according to the clinical outcomes, although there was no significant difference between the experimental and control groups in terms of *p*-value [15]. The limitation of the present study is that there was a small number of participants, therefore further clinical studies are required with increased number of patients to confirm these clinical effects of 915 nm LLLT on mucosal wound healing.

It has been well reported that diode laser with a wavelength in the range of 655–980 nm is able to accelerate wound healing by promoting angiogenesis and release of growth factors. The results of the present study were similar to those of the previous studies, which reported that 685 nm LLLT appears to promote healing [32], and 588 nm LLLT enhanced epithelialization following gingivectomy [33]. Compared to 685 or 588 nm diode lasers, the 915 nm laser is less absorbed by pigmented tissues, so there would be low risk of heat production and thermal damage. For these reasons, a diode laser with 915 nm wavelength was used after surgical incision on intraoral mucosa. In the present study, there were no specific postoperative complications in the LLLT group except one patient who showed similar levels of postoperative pain and swelling on both control and LLLT sides. For the postoperative medication in the protocol, antibiotics and non-steroidal anti-inflammatory drugs (NSAIDs) were prescribed on the operation day only, and the patients were not allowed to take any non-prescribed medication. Soon after further medication, the symptoms of the patient with swelling on both sides were subsided. Therefore, the mild symptoms were not seemed to be induced by laser irradiation on the LLLT side but by transient alveolar osteitis related to low immunity of the patient, which is the most frequent postoperative complication observed in about 36% of extraction patients [11].

Although a number of recent clinical studies are reporting favorable effects of LLLT on intraoral tissues, it has not yet been clearly demonstrated that LLLT is superior to the conventional therapy [34]. In contrast to the efficacies of LLLT with respect to promoting wound healing in the medical fields, those in the intraoral areas are still controversial. First, most of the studies about intraoral areas used different parameters and methodologies of laser application such as output power, irradiation dosage and time, pulse rate, wavelength of laser, and application methods which could cause discrepancy of the results. For example, LLLT by extraoral approach shows more positive effects on reducing postoperative complications compared to intraoral LLLT [35]. Moreover, despite the fact that many studies showed the effect of promoting wound healing by LLLT, lasers could lead to inhibition of wound healing under other parameters according to the Arndt–Schulz rule [36]. As a result, laser therapy for treating acute problems such as postoperative swelling and inflammatory symptoms could be used until symptoms are cured, while in case of chronic problems, it should not be used for longer than two weeks postoperatively [37]. It has been demonstrated that 980 nm LLLT (continuous mode, at 0.3 W, 1 min × 3, 54 J) is useful for the reduction of postoperative swelling and trismus after extraction [17]. Mozzati et al. have shown that 904 nm laser therapy (0.2 W, 15 min, 180 J, 180 J/cm^2^) prevents the increase of inflammatory cytokines such as interleukins and cyclooxygenase-2 at seven days after extraction in the clinical study with a biopsy of human specimen [30]. In the present study, 187.5 J/cm^2^ in fluency of LLLT was used in the clinical study by considering the previous studies. However, the presence of blood from the extraction socket acts as another considering factor for the effect and safety of LLLT in oral cavity [38]. The blood could considerably increase the absorption of laser energy and may increase the risk of thermal damage. Further extensive clinical studies are required with an increased number of patients to clarify the definite roles of 915 nm LLLT on mucosal wound healing and to find proper irradiation conditions and delivery methods.

## 5. Conclusions

In conclusion, LLLT with 915 nm wavelength diode laser promoted intraoral mucosal wound healing histologically, and repeated irradiation of 915 nm laser showed better results than a single irradiation in the animal study. Although there were no significant statistical differences between the LLLT and control groups in all inflammatory parameters in the exploratory clinical study, 915 nm laser, as an adjunct to mucosal wound healing, is safe and decreases the occurrence of adverse events and immediate postoperative discomforts clinically. Therefore, this study has demonstrated that 915 nm LLLT is useful for the reduction of immediate postoperative complications after intraoral surgeries including surgical extraction, which usually involves postoperative pain, trismus, and swelling due to the inflammatory process initiated by surgical trauma. It could be safely applied as an auxiliary treatment for mucosal wound healing and inflammation.

## Figures and Tables

**Figure 1 medicina-55-00405-f001:**
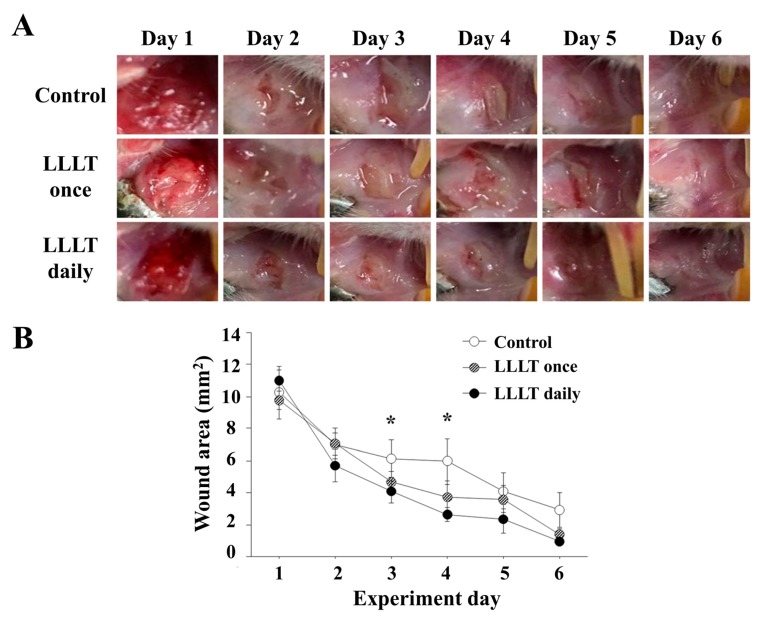
Progress of wound healing. (**A**) Clinical photographs showing the wound healing progress across the control, LLLT once and LLLT daily groups. All of the surgical wounds recovered within 7 days clinically. (**B**) Graph of wound area measured with Image J software. On days 3 and 4, there were significant differences between the control group and LLLT daily group (* Control vs. LLLT daily, *p* < 0.05).

**Figure 2 medicina-55-00405-f002:**
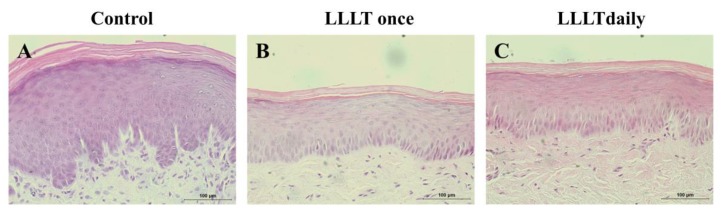
Histological analysis of epithelial thickness of the wounds after 7 days of healing periods. (**A**) Control group. The epithelial layers of the wound site were thicker than those of normal tissue. (**B**) LLLT once group. The thickness of the epithelial layers of the wound site showed still irregular manner similar to the control group. (**C**) LLLT daily group. Scale bar, 100 μm. Compared with the other two groups, the LLLT daily group showed thin uniform epithelial layers of the wound site similar to that of normal tissues.

**Figure 3 medicina-55-00405-f003:**
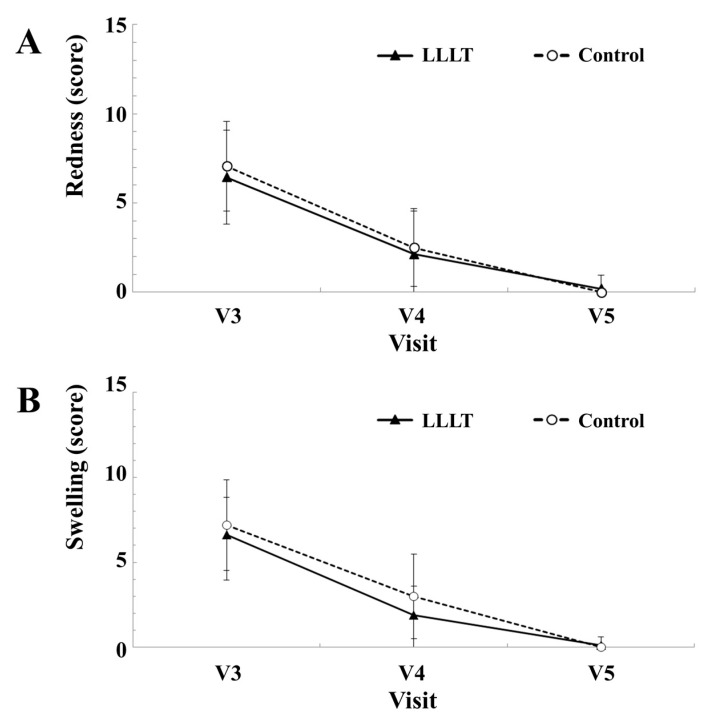
Comparison between experimental LLLT group and control placebo group. (**A**) Average scores for redness at each visit. (**B**) Average scores for swelling at each visit.

**Figure 4 medicina-55-00405-f004:**
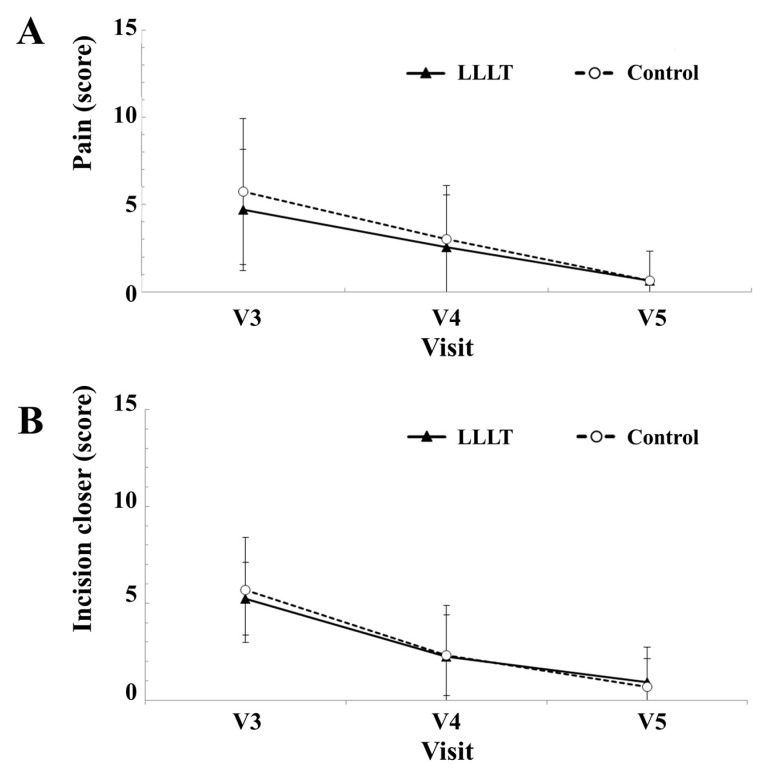
Comparison between experimental LLLT group and control placebo group. (**A**) Average scores for pain at each visit. (**B**) Average scores for incision wound closure at each visit.

**Figure 5 medicina-55-00405-f005:**
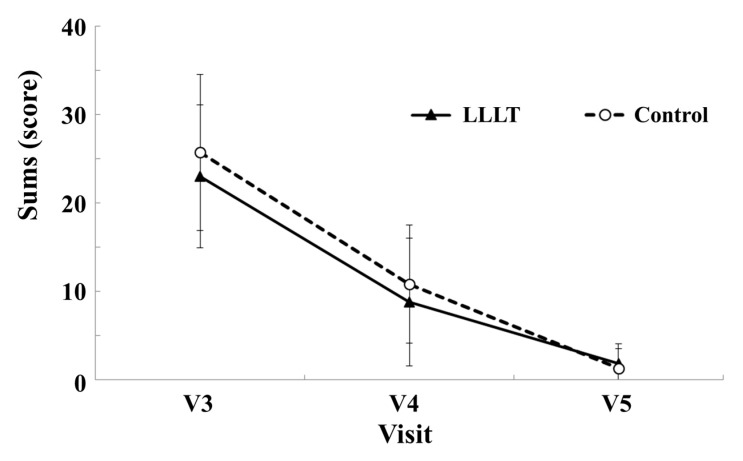
Sums of scores for all four parameters (redness, swelling, pain, and dehiscence of the wound). Comparison between experimental LLLT group and control placebo group.

**Table 1 medicina-55-00405-t001:** The experimental animals grouping. Thirty-six rats were used in total for each group with 6 rats sampled daily for 6 days. Each group had 6 animals per day for 6 days, total 108 rats. LLLT, low-level laser therapy.

Group	Control (n = 36)	LLLT Once (n = 36)	LLLT Daily (n = 36)
1 day	6	6	6
2 day	6	6	6
3 day	6	6	6
4 day	6	6	6
5 day	6	6	6
6 day	6	6	6

**Table 2 medicina-55-00405-t002:** Local clinical symptoms observed in the clinical study and the scoring points. The points were used to evaluate the presence of those symptoms in all patients.

Parameters of Local Clinical Symptoms	Scoring Points
Redness	0: Complete absence of symptom
Swelling	5: Mild
Pain	10: Moderate
Incision wound closure	15: Severe

**Table 3 medicina-55-00405-t003:** Comparison of adverse events in both LLLT and control groups.

Adverse Event	Control (n = 16) N (%)	LLLT (n = 16) N (%)	*p*-Value ^1)^
Yes	3 (18.75)	1 (6.25)	0.5996
No	13 (81.25)	15 (93.75)	

LLLT, low-level laser therapy; ^1)^ Fisher’s exact test.

**Table 4 medicina-55-00405-t004:** Details of adverse event cases in both LLLT and control groups.

System Organ Class *	Preferred Term	Control (n = 16) N (%)	LLLT (n = 16) N (%)
Gastrointestinal disorders	postoperative pain and swelling	2 (12.50)	1 (6.25)
Infections and infestations	postoperative infection	1 (6.25)	

LLLT, low-level laser therapy; * Duplicate counting, MedDRA 19.1.

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
