# Peer review of "Effectiveness of Low-Level Laser Therapy with a 915 Nm Wavelength Diode Laser on the Healing of Intraoral Mucosal Wound: An Animal Study and a Double-Blind Randomized Clinical Trial"

_1010-660X, 2019, doi:10.3390/medicina55080405_

Round 1

Reviewer 1 Report

Please check Page 5/12, lines 173>184. 

The text is not related to the "Statistical Analysis" description (maybe instructions for authors?).

There is a discrepancy between the number of rats used for the animal study: on line 23 (abstract) and line 74 (animal study) a total of 128 rats is reported, while on note of Tab. 1 a total of 108 rats is reported, as resulting from the sum of 3 groups of 36 rats each.

Author Response

Dear Anonymous Reviewer #1

We would like to thank the reviewer for careful and thorough reading of this manuscript and for the thoughtful comments and constructive suggestions, which help to improve the quality of this manuscript.

The following is a point-by-point response to the referees’ comments

1. Please check Page 5/12, lines 173>184. The text is not related to the "Statistical Analysis" description (maybe instructions for authors?).

Response: Thank you for your kind comment. As you suggested, we have revised those paragraphs in the article (page 8 lines 175-186).

2. There is a discrepancy between the number of rats used for the animal study: on line 23 (abstract) and line 74 (animal study) a total of 128 rats is reported, while on note of Tab. 1 a total of 108 rats is reported, as resulting from the sum of 3 groups of 36 rats each.

Response: Thank you for your kind comment. As you suggested, we have changed the number of rat from 128 to 108 in the abstract (typo-error) and materials and methods parts. (Page 1 line 22, page 2 line 77)

Reviewer 2 Report

The manuscript topic is actual and the paper has merit. It could be attractive, adequate and interesting for the MEDICINA MDPI journal readers. However there are some point that authors should address in order to have a final more complete paper. Authors should underline the limitation of the value of the study, and the clinical and surgical implication of the presented study should be added. At this stage the paper seems to be directed to not clinical or surgeons readers. Please emphasize the clinical application of the study.
The limitation of an "animal study" should be underlined and need to be synthesized in a paragraph. 
....animal studies will only become more valid predictors of human reactions to exposures and treatments if there is substantial improvement in both their scientific methods as well as in more systematic review of the animal literature as it evolves. Systematic reviews of animal research, if they are used to inform the design of clinical trials, particularly with respect to appropriate drug dose, timing and other crucial aspects of the drug regimen, will further improve the predictability of animal research in human clinical trials....

Introduction section should highlights the clinical rationale of this paper. Otherwise the study seems to be directed to just scientist or researcher and not to surgeons. 

References are inadequate. Introduction section is poor. Some more references may be increase the valute of the discussion

Fiorillo, L.; Cervino, G.; Herford, A.S.; Lauritano, F.; D’Amico, C.; Lo Giudice, R.; Laino, L.; Troiano, G.; Crimi, S.; Cicciù, M. Interferon Crevicular Fluid Profile and Correlation with Periodontal Disease and Wound Healing: A Systemic Review of Recent Data. Int. J. Mol. Sci. 201819, 1908.

Fiorillo, L. Chlorhexidine Gel Use in the Oral District: A Systematic Review. Gels 20195, 31.

Author Response

Dear Anonymous Reviewer #2

We would like to thank the reviewer for careful and thorough reading of this manuscript and for the thoughtful comments and constructive suggestions, which help to improve the quality of this manuscript.

The following is a point-by-point response to the referees’ comments

1. Authors should underline the limitation of the value of the study, and the clinical and surgical implication of the presented study should be added.

Response: This is a good comment to lead our paper more scientific. Following your comment, we have revised and added the expression in the discussion section as followings:

(Page 9 lines 321-324) “The limitation of the present study is that there was a small number of participants, therefore further clinical studies are required with increased number of patients to confirm these clinical effects of 915 nm LLLT on mucosal wound healing.”

(Page 10 lines 359-364) “However, the presence of blood from the extraction socket acts as another considering factor for the effect and safety of LLLT in the oral cavity. The blood could considerably increase the absorption of laser energy and may increase the risk of thermal damage. Further extensive clinical studies are required with increased number of patients to clarify the definite roles of 915 nm LLLT on mucosal wound healing and to find proper irradiation conditions and delivery methods.”

2. At this stage the paper seems to be directed to not clinical or surgeons readers. Please emphasize the clinical application of the study.

Response: Thank you for your valuable comment. As you suggested, we have added several sentences in conclusion section as followings:

(Page 10 lines 358-363) “Therefore, this study has demonstrated that 915 nm LLLT is useful for the reduction of immediate postoperative complications after intraoral surgeries including surgical extraction, which usually involves severe postoperative pain, trismus and swelling due to the inflammatory process initiated by surgical trauma. It could be safely applied as an auxiliary treatment for mucosal wound healing and inflammation.”

3. The limitation of an "animal study" should be underlined and need to be synthesized in a paragraph. 
....animal studies will only become more valid predictors of human reactions to exposures and treatments if there is substantial improvement in both their scientific methods as well as in more systematic review of the animal literature as it evolves. Systematic reviews of animal research, if they are used to inform the design of clinical trials, particularly with respect to appropriate drug dose, timing and other crucial aspects of the drug regimen, will further improve the predictability of animal research in human clinical trials....

Response: Thank you for your valuable comment. As you suggested, we have synthesized a paragraph about the limitation of an animal study in the discussion section as followings:

(Page 9 lines 306-310) “The limitation of the animal study is that the wounds were not identical between animal and clinical studies. The intraoral wound of animal was abrasion wound whereas human wound was mucoperiosteal incisional wound. Furthermore, intraoral mucosal thickness and healing potential are different between those two species. Therefore, we used some different parameters of 915 nm laser irradiation in animal and clinical studies because the wounds were not identical.”

4. Introduction section should highlights the clinical rationale of this paper. Otherwise the study seems to be directed to just scientist or researcher and not to surgeons. 

Response: Thank you for your kind comment. As you suggested, we have added the expression to highlight the clinical rationale of this paper in introduction section as followings:

(Page 1 lines 40-44) “Surgical extraction of third molar is one of the most common oral surgery in dentistry. After extraction, a traumatic inflammatory response occurs along the mucosal wound with pain and facial swelling. The mucosal wound is remained under high risk of infection in oral environment. To reduce the postoperative complication and promote healing of the mucosal wounds, application of additional methods could be considered.”

5. References are inadequate. Introduction section is poor. Some more references may be increase the value of the discussion

Response: Special thanks to your great suggestion. Following your comment, the reference format has been corrected. We have revised the introduction section and added more references:

1. Sato, F.R.; Asprino, L.; de Araujo, D.E.; de Moraes, M. Short-term outcome of postoperative patient recovery perception after surgical removal of third molars. J Oral Maxillofac Surg. 2009, 67, 1083-1091.

2. Kim, K.; Brar, P.; Jakubowski, J.; Kaltman, S.; Lopez, E. The use of corticosteroids and nonsteroidal antiinflammatory medication for the management of pain and inflammation after third molar surgery: a review of the literature. Oral Surg Oral Med Oral Pathol Oral Radiol Endod. 2009, 107, 630-640.

20. Farhadi, F.; Eslami, H.; Majidi, A.; Fakhrzadeh, V.; Ghanizadeh, M.; KhademNeghad, S. Evaluation of adjunctive effect of low-level laser Therapy on pain, swelling and trismus after surgical removal of impacted lower third molar: A double blind randomized clinical trial. Laser Ther. 2017, 26, 181-187.

22. de Matos Brunelli Braghin, R.; Libardi, E.C.; Junqueira, C.; Rodrigues, N.C.; Nogueira-Barbosa, M.H.; Renno, A.C.M.; Carvalho de Abreu, D.C. The effect of low-level laser therapy and physical exercise on pain, stiffness, function, and spatiotemporal gait variables in subjects with bilateral knee osteoarthritis: a blind randomized clinical trial. Disabil Rehabil. 2018, 1-8.

23. Wanitphakdeedecha, R.; Iamphonrat, T.; Phothong, W.; Eimpunth, S.; Manuskiatti, W. Local and systemic effects of low-level light therapy with light-emitting diodes to improve erythema after fractional ablative skin resurfacing: a controlled study. Lasers Med Sci. 2019, 34, 343-351.

24. Mai-Yi Fan, S.; Cheng, Y.P.; Lee, M.Y.; Lin, S.J.; Chiu, H.Y. Efficacy and Safety of a Low-Level Light Therapy for Androgenetic Alopecia: A 24-Week, Randomized, Double-Blind, Self-Comparison, Sham Device-Controlled Trial. Dermatol Surg. 2018, 44, 1411-1420.

25. Rodrigues Neves, C.; Buskermolen, J.; Roffel, S.; Waaijman, T.; Thon, M.; Veerman, E.; Gibbs, S. Human saliva stimulates skin and oral wound healing in vitro. J Tissue Eng Regen Med. 2019, 13, 1079-1092.

27. Menon, R.K.; Gopinath, D.; Li, K.Y.; Leung, Y.Y.; Botelho, M.G. Does the use of amoxicillin/amoxicillin-clavulanic acid in third molar surgery reduce the risk of postoperative infection? A systematic review with meta-analysis. Int J Oral Maxillofac Surg. 2019, 48, 263-273.

34. Weber, J.B.; Camilotti, R.S.; Ponte, M.E. Efficacy of laser therapy in the management of bisphosphonate-related osteonecrosis of the jaw (BRONJ): a systematic review. Lasers Med Sci. 2016, 31, 1261-1272.